# Text-DiFuse: An Interactive Multi-Modal Image Fusion Framework based on Text-modulated Diffusion Model

**Hao Zhang**,[*] **Lei Cao**,[*] **Jaiyi Ma**[†]
Electronic Information School
Wuhan University
Wuhan, China
`{zhpersonalbox, jyma2010}@gmail.com, whu.caolei@whu.edu.cn`

## Abstract

Existing multi-modal image fusion methods fail to address the compound degradations presented in source images, resulting in fusion images plagued by noise, color bias, improper exposure, *etc*. Additionally, these methods often overlook the specificity of foreground objects, weakening the salience of the objects of interest within the fused images. To address these challenges, this study proposes a novel interactive multi-modal image fusion framework based on the text-modulated diffusion model, called Text-DiFuse. First, this framework integrates feature-level information integration into the diffusion process, allowing adaptive degradation removal and multi-modal information fusion. This is the first attempt to deeply and explicitly embed information fusion within the diffusion process, effectively addressing compound degradation in image fusion. Second, by embedding the combination of the text and zero-shot location model into the diffusion fusion process, a text-controlled fusion re-modulation strategy is developed. This enables user-customized text control to improve fusion performance and highlight foreground objects in the fused images. Extensive experiments on diverse public datasets show that our Text-DiFuse achieves state-of-the-art fusion performance across various scenarios with complex degradation. Moreover, the semantic segmentation experiment validates the significant enhancement in semantic performance achieved by our text-controlled fusion re-modulation strategy. The code is publicly available at `https://github.com/Leiii-Cao/Text-DiFuse`.

## 1 Introduction

Due to constraints in imaging principles and hardware technology, single-modal images fall short of accurately and comprehensively describing scenes, thereby limiting their utility in subsequent tasks. Hence, image fusion technology emerges as essential in this context [60, 28]. It aims to integrate useful information from multi-modal images, producing high-quality visual results that enhance both human and machine perception of scenes. Currently, image fusion technology has been integrated into various tasks, significantly advancing performance in related fields such as autonomous driving [47], intelligent security [57, 25], and disease diagnosis [14].

Over recent decades, rapid advancements in deep learning have propelled significant progress in image fusion. Deep learning-based methods have surpassed traditional approaches in fusion performance by a considerable margin. In the historical context, the evolution of image fusion closely aligns with the

---

[*]Equal Contribution
[†]Corresponding author

38th Conference on Neural Information Processing Systems (NeurIPS 2024).

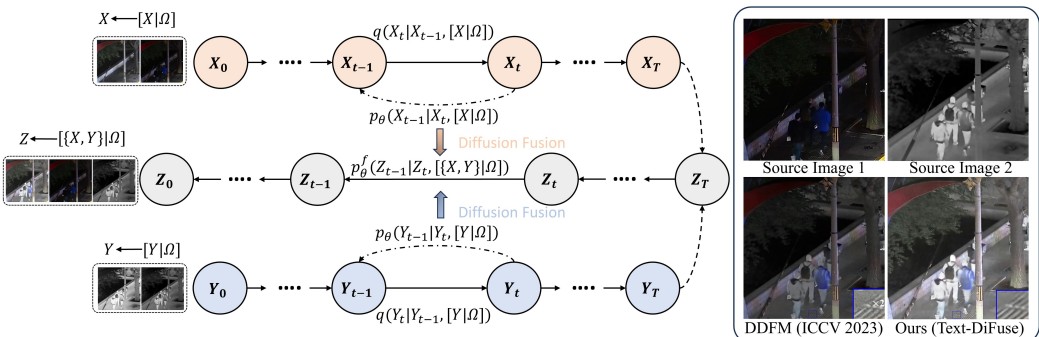

Figure 1: Our proposed explicit coupling paradigm of multi-modal information fusion and diffusion.

advancements in network paradigms. Autoencoders (AE) [19, 20], convolutional neural networks (CNN) [54, 49], generative adversarial networks (GAN) [29, 31], and Transformers [32, 44] represent a coherent axis of progress in image fusion. This phenomenon arises because there is no ground truth to supervise fusion learning. Therefore, fusion performance depends heavily on the continuous enhancement of the feature expression potential of neural network paradigms [55].

However, these methods falter in scenes with degradation, especially composite degradation, which we refer to as the **composite degradation challenge**. Essentially, current methods *prioritize multi-modal information integration without considering effective information restoration from degradation* [62, 26]. Last few years, the emergence of diffusion models has impressed many with their remarkable performance in visual restoration tasks [8, 16]. This prompts a natural question: Can diffusion models be utilized to tackle the challenge of multi-modal image fusion in scenes with complex degradation? According to the research status of diffusion model-based image fusion, two factors have hindered the implementation of this intuitive idea. First, visual restoration using diffusion models requires pairs of degraded and clean images, but in image fusion tasks, there is no clean fused image as ground truth due to the unsupervised nature of the task. Second, breaking through the paradigm that explicitly couples information fusion and diffusion is yet to be achieved.

Moreover, current fusion methods fail to account for the specificity of objects in the scene (*e.g.*, pedestrians, vehicles), applying the same fusion rules indiscriminately to both foreground and background. This lack of differentiation, termed the **under-customization objects limitation**, is unreasonable and may compromise the delineation of crucial objects [53, 64, 51]. Undoubtedly, maintaining the salience of foreground objects is crucial to satisfy both human and machine interest in them. This necessitates fusion models to possess the capability of interacting with users, achieving a "what you are interested in is what you get" approach.

To address the challenges of **composite degradation** and **under-customization objects** in multi-modal image fusion, we propose a novel interactive multi-modal image fusion framework based on the text-modulated diffusion model (Text-DiFuse). On the one hand, Text-DiFuse customizes a new explicit coupling paradigm of multi-modal information fusion and diffusion, eliminating complex degradation like color casts, noise, and improper lighting, as shown in Fig. 1. Specifically, it first applies independent conditional diffusion to data with compounded degradation, enabling degradation removal priors to be embedded into the encoder-decoder network. A fusion control module (FCM) is then embedded between the encoder and decoder to manage the integration of multi-modal features. This involves fusing multiple diffusion processes at the feature level, continuously aggregating multi-modal information while removing degradation during T-step sampling. To our knowledge, this is the first time information fusion is deeply and explicitly embedded in the diffusion process, effectively addressing compound degradation in image fusion tasks. On the other hand, to interactively enhance focus on objects of interest during diffusion fusion, we design a text-controlled fusion re-modulation strategy. This strategy incorporates text and a zero-shot location model to identify the objects of interest, thereby performing secondary modulation with the built-in prior to enhance their saliency. Thus, both the visual quality and semantic attributes of the fused image are significantly improved.

In summary, we make the following contributions:

• We propose a novel explicit coupling paradigm of information fusion and diffusion, solving the compound degradation challenge in the task of multi-modal image fusion.

• A text-controlled fusion re-modulation strategy is designed, allowing users to customize fusion rules with language to enhance the salience of objects of interest. This interactively improves the visual quality and semantic attributes of fused images.

• We evaluate our Text-DiFuse on extensive datasets and verify its advantages over state-of-the-art methods in terms of degradation robustness, generalization ability, and semantic properties.

## 2 Related Work

**Deep Multi-modal Image Fusion.** As mentioned earlier, the progress in deep multi-modal image fusion is closely tied to updates in neural network paradigms. Initially, AE-based fusion methods [19, 20] utilize pre-trained encoders and decoders alongside hand-crafted fusion rules, resulting in performance bottlenecks. Subsequent methods introduce CNN [6, 5] and Transformer [37, 58] for end-to-end fusion guided by specific unsupervised loss, yielding improved performance. The introduction of GAN is groundbreaking due to their inherently unsupervised nature, enabling the preservation of important multi-modal features [29, 56]. However, the instability of the adversarial game often leads to non-equilibrium appearances in fused images [30]. Furthermore, the diffusion model is highly anticipated for solving image fusion and is used in two main ways: injecting features into CNNs for separate fusion and diffusion [52], or treating multi-modal images as conditions for implicit fusion [63]. However, both methods fail to utilize the diffusion model's degradation removal capabilities and struggle with complex degradation. In contrast, our Text-DiFuse embeds feature fusion into the diffusion process, ensuring robustness and aggregation of multi-modal information. Additionally, we use text combined with a zero-shot location model for user-customized fusion, enhancing object salience.

**Diffusion Model.** The impressive performance of the diffusion model [41, 13] makes it top-notch in visual generation. It constructs a Markov chain by progressively adding noise forward, and then estimates the underlying data distribution and uses inverse sampling to generate images. This natural property of degradation removal has made the diffusion model excel in visual restoration tasks [46, 61]. However, the practical application of the diffusion model is hindered by its slow T-step continuous sampling. Recent efforts have focused on enhancing sampling efficiency and sample quality [50]. For example, DDIM [42] extends the original denoising diffusion probability model to non-Markovian scenarios, requiring only discrete time steps during sampling to reduce costs. Furthermore, iDDPM [35] introduces an enhanced denoising diffusion probability model, parameterizing backward variance through linear interpolation and training with mixed objectives to acquire knowledge of backward variance. This approach increases log-likelihood and accelerates sampling rates without compromising sample integrity. Therefore, our method employs iDDPM to expedite sampling while upholding the quality of fused images.

**Zero-shot Location.** Establishing connections between unseen and seen categories using semantic information [18], zero-shot location models [27, 33] can understand unseen images to identify and locate designated objects. Representative zero-shot location models include GLIP [22], OWL-VIT [33], and Grounding DINO [27]. Additionally, methods like DiffSeg [40], PADing [12], and SAM [17] achieve finer pixel-level object localization. These powerful zero-shot location techniques provide a solid foundation for the implementation of our text-controlled fusion re-modulation.

## 3 Methodology

### 3.1 Problem Statement and Modeling

Let us formally define the research problem of this work: achieving multi-modal image fusion under degraded scenes while supporting text-controlled fusion re-modulation of objects of interest. The multi-modal image pair captured under degraded conditions is formulated $[\{X, Y\}|\Omega]$, in which $\{X, Y\}$ denotes clean multi-modal images (*e.g.*, infrared and visible images), and $\Omega$ indicates composite degradation (*e.g.*, color casts, noise, and improper lighting). We aim to process degraded multi-modal images to obtain a clean fused image: $Z = \Gamma([\{X, Y\}|\Omega])$. The function $\Gamma$ must handle two tasks: degradation removal $R$ and information fusion $F$. There are two routes: concatenation ($\Gamma = R + F$) and coupling ($\Gamma = R \uplus F$). Concatenation overlooks the intrinsic connection between degradation removal and fusion, leading to limited performance (see comparative experiments).

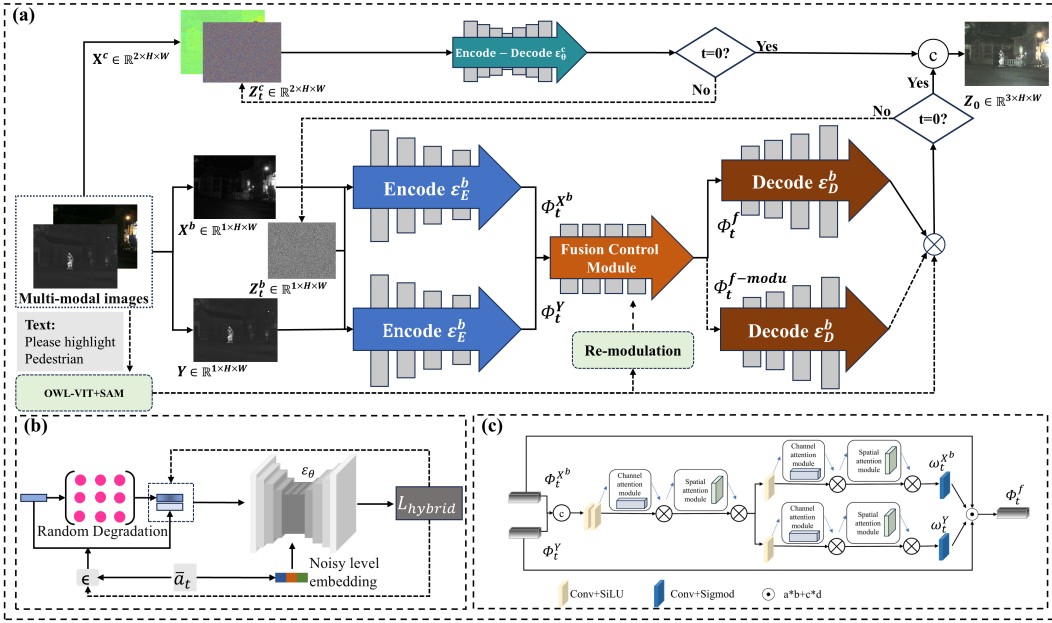

Figure 2: The pipeline of our Text-DiFuse. (a) The text-controlled diffusion fusion process; (b) training diffusion model for degradation removal; (c) the detailed structure of fusion control module.

Therefore, we choose the coupling route and introduce the diffusion model for degradation removal. Then, the key question becomes how to integrate the diffusion model with information fusion. Notably, the diffusion operates as a continuous process with multi-step sampling, making it difficult to incorporate fusion. To address this, we propose a novel **explicit coupling paradigm of information fusion and diffusion**, as illustrated in Fig. 2 (a). Specifically, the diffusion model is initially trained on data with compound degradation, incorporating the degradation removal prior into the encoder $\varepsilon_E^b$ and decoder $\varepsilon_D^b$. During T-step reverse sampling, the multi-modal encoded features are continuously passed to the FCM for fusion, aiding in the reconstruction of the final fused image by the decoder. This essentially consolidates multiple diffusion processes into a single one, effectively integrating degradation removal and information fusion. However, the above methodology has not yet addressed the re-modulation of objects of interest in image fusion. To this end, we further develop a **text-controlled fusion re-modulation strategy** to highlight the objects of interest, thereby enhancing subsequent semantic decision performance. This strategy can be formulated as $Z = \Gamma([\{X, Y\}|\Omega], L)$, where $L$ represents the user-defined language command. Specifically, we utilize text combined with the zero-shot location module to identify and locate the objects of interest. This knowledge triggers the re-modulation of diffusion fusion to enhance the saliency of the objects with in-built contrast-enhancement prior, thereby significantly improving perceptual quality for both humans and machines.

## 3.2 Explicit Coupling Paradigm of Information Fusion and Diffusion

**Diffusion for Degradation Removal.** Embedding the degradation removal prior into the encoder-decoder network of the diffusion model forms the foundation for diffusion fusion. We consider three primary types of degradation: color casts, noise, and improper lighting. They cover both nighttime and daytime negative imaging conditions and can be considered comprehensive to a certain extent. In our model, we separate brightness and chrominance components, and perform independent diffusion for them. Here, we describe and represent these two diffusion processes consistently. For clean components $s$ (brightness or chrominance), the corresponding degraded versions $\Omega(s)$ involve composite degradation like color casts, noise, and improper lighting. These degraded components are fed into the encoder-decoder network as conditions. As shown in Fig. 2 (b), in the forward diffusion process, the original clean component $s$, denoted as $s_0$ at step 0, is progressively added with Gaussian noise over $T$ steps to obtain $s_T$. In the reverse sampling process, the encoder-decoder network is guided to estimate the mean $\mu_\theta(s_t, \Omega(s), t)$ and variance $\sum_\theta(s_t, \Omega(s), t)$ of $s_{t-1}$,

progressively approaching the clean component with the conditions $\Omega(s)$. Drawing upon iDDPM [35], the optimization can be defined as:

$$\nabla_\theta \parallel \epsilon_t - \epsilon_\theta(s_t, \Omega(s), t) \parallel^2 + \lambda D_{\mathrm{KL}}(q(s_{t-1}|s_t, s_0, \Omega(s))||p_\theta(s_{t-1}|s_t, \Omega(s))), t > 1 \qquad (1a)$$

$$\nabla_\theta \parallel \epsilon_t - \epsilon_\theta(s_t, \Omega(s), t) \parallel^2 - \lambda \log p_\theta(s_0|s_1, \Omega(s)), t = 1 \qquad (1b)$$

where $\nabla_\theta$ means optimization by gradient descent, $\epsilon_t$ denotes the added noise in the forward diffusion process, $D_{\mathrm{KL}}$ is the regularization term based on KL divergence, and $q$ and $p_\theta$ represent the prior and posterior probability distributions, respectively. $\epsilon_\theta$ is the noise predictor.

**Fusion Control Module for Information Fusion.**   For information fusion, we design an FCM to aggregate encoded features during the diffusion process, as shown in Fig. 2 (c). It primarily consists of convolution layers with CBAM [45]. In them, integrating spatial and channel attention mechanisms helps perceive the importance of multi-modal features on a wider scale, promoting rational feature fusion. To reduce the solution space, FCM generates weight coefficients for fusion rather than directly predicting fused features, allowing faster convergence in multi-step sampling of the diffusion process.

**Diffusion Fusion.**   Everything is ready, and now we can seamlessly integrate information fusion with diffusion, termed diffusion fusion. Given the degraded multi-modal image pairs $[\{X, Y\}|\Omega]$, we assume $X$ is a color image and $Y$ is a grayscale image. This assumption aligns with most multi-modal image fusion scenarios, such as visible and infrared image fusion, and MRI and PET image fusion. By separating components, we obtain the brightness component $[X^b|\Omega]$ and chrominance component $[X^c|\Omega]$. First, using the trained diffusion model $\varepsilon_\theta^c$, we remove the degradation in the chrominance component $[X^c|\Omega]$, obtaining a clean and reasonable chrominance component $X_0^c$.

Then, the processing of paired $[\{X^b, Y\}|\Omega]$ involves diffusion fusion, which achieves simultaneous degradation removal and information fusion. Specifically, given randomly sampled Gaussian noise $Z_T^b \sim N(0, I)$, $[\{X^b, Y\}|\Omega]$ are regarded as the condition input to the shared encoder $\varepsilon_E^b$ in another diffusion model, obtaining features $[\{\Phi_t^{X^b}, \Phi_t^Y\}|\Omega]$ at step $t$:

$$[\{\Phi_t^{X^b}, \Phi_t^Y\}|\Omega] = \varepsilon_E^b(Z_t^b, [\{X^b, Y\}|\Omega], t), t \in \{T, \cdots 0\}. \qquad (2)$$

The FCM generates weight coefficients $\{\omega_t^{X^b}, \omega_t^Y\}$ for fusing these multi-modal features:

$$[\Phi_t^f|\Omega] = [\{\Phi_t^{X^b}, \Phi_t^Y\}|\Omega] \odot \{\omega_t^{X^b}, \omega_t^Y\}, \qquad (3)$$

where $[\Phi_t^f|\Omega]$ is the fused feature with residual degradation at step $t$, and $\odot$ denotes the Hadamard product. Subsequently, the fused feature is fed into the decoder $\varepsilon_D^b$ to predict the contained noise $\epsilon_\theta(t)$ at step $t$, and the relevant variable $\upsilon_\theta(t)$ for learning the variance:

$$\epsilon_\theta(t), \upsilon_\theta(t) = \varepsilon_D^b([\Phi_t^f|\Omega], t). \qquad (4)$$

Then, the mean and variance of $Z_{t-1}^b$ can be obtained according to:

$$\mu_\theta(Z_t^b, [\{X^b, Y\}|\Omega], t) = \frac{1}{\sqrt{\alpha_t}}(Z_t^b - \frac{1-\alpha_t}{\sqrt{1-\overline{\alpha}_t}}\epsilon_\theta(t)), \qquad (5)$$

$$\sum\nolimits_\theta(Z_t^b, [\{X^b, Y\}|\Omega], t) = exp(\upsilon_\theta(t) \log \beta_t + (1 - \upsilon_\theta(t)) \log \tilde{\beta}_t), \qquad (6)$$

where $\beta_t$ represents the variance associated with the forward diffusion process, using the notation $\alpha_t = 1 - \beta_t$ and $\overline{\alpha}_t = \prod_{s=0}^t \alpha_s$ . Additionally, we parameterize the variance between $\beta_t$ and $\tilde{\beta}$ in the logarithmic domain using the technique of iDDPM [35], where $\tilde{\beta} = \frac{1-\overline{\alpha}_{t-1}}{1-\overline{\alpha}_t}\beta_t$. Then, $Z_{t-1}^b$ can be computed according to:

$$Z_{t-1}^b = \mu_\theta(Z_t^b, [\{X^b, Y\}|\Omega], t) + \sqrt{\sum\nolimits_\theta(Z_t^b, [\{X^b, Y\}|\Omega], t)} \cdot z, \qquad (7)$$

where $z$ denotes the randomly sampled Gaussian noise $z \sim N(0, I)$ when $t > 1$, otherwise $z = 0$. According to Eqs. (4)-(7), each sample will derive an $\hat{Z}_0^b$:

$$\hat{Z}_0^b(Z_t^b, \epsilon_\theta(t)) = \frac{Z_t^b - \sqrt{1-\overline{\alpha}}\epsilon_\theta(t)}{\sqrt{\overline{\alpha}}}. \qquad (8)$$

Notably, $\hat{Z}_0^b(Z_t^b, \epsilon_\theta(t))$ indicates the corresponding fake final fused image that is derived from the results of any step of sampling. Therefore, we construct constraints to guide the FCM in retaining beneficial information during the diffusion fusion process. Considering pixel intensity and gradient as two basic elements that describe images, we specify intensity loss $\mathcal{L}_{int}$ and gradient loss $\mathcal{L}_{grad}$ to emphasize the preservation of significant contrast and rich texture:

$$\mathcal{L}_{int} = \| \, |\hat{Z}_0^b(Z_t^b, \epsilon_\theta(t))| - \max\{|X^b|, |Y|\} \, \|, \tag{9}$$

$$\mathcal{L}_{grad} = \| \, \nabla\hat{Z}_0^b(Z_t^b, \epsilon_\theta(t)) - \max\{\nabla X^b, \nabla Y\} \, \|, \tag{10}$$

where $\max$ is the maximum function, $\nabla$ is the Sobel gradient operator, $X^b$ and $Y$ are clean source components after diffusion. The total loss is summarized as:

$$\mathcal{L}_{diff-fusion} = \gamma_{int}\mathcal{L}_{int} + \gamma_{grad}\mathcal{L}_{grad}, \tag{11}$$

where $\gamma_{int}$ and $\gamma_{grad}$ control the balance of these terms, set to $1$ and $0.2$, respectively. After optimization, we obtain the clean fused brightness component $Z^b = Z_0^b$. The purified chrominance component $X^c$ is used as the fused image's chrominance component: $Z^c = X_0^c$. Finally, stitching $Z^b$ and $Z^c$ yields the final fused image $Z$ with accurate colors, minimal noise, and proper lighting. Through these designs, information fusion and diffusion have been fully and explicitly coupled, achieving multi-modal image fusion while removing compound degradation.

### 3.3 Text-controlled Fusion Re-modulation Strategy

The above diffusion fusion constitutes the **basic version** of our method. Now, we aim to expand it into a **modulatable version**, allowing users to re-modulate the fusion process based on personalized needs, enhancing the perception of objects of interest. Firstly, we use state-of-the-art zero-shot localization models to identify and locate objects of interest based on text commands. Specifically, we introduce OWL-VIT [33] for detecting objects of interest with open-word input. Then, SAM [17] provides pixel-level positioning of these objects, obtaining the mask $M$. Subsequently, $M$ is fed into the re-modulation block to generate fusion modulation coefficients $\{\kappa^{X^b}, \kappa^Y\}$. This block incorporates a built-in contrast-enhancement prior, aiming to maximize the contrast between the object area and the background in the fused image, thus improving the salience of the objects. Consequently, the multi-modal feature fusion in the diffusion process changes from Eq. (3) to:

$$[\Phi_t^{f-modu}|\Omega] = [\{\Phi_t^{X^b}, \Phi_t^Y\}|\Omega] \odot \{\omega_t^{X^b}, \omega_t^Y\} \odot \{\kappa^{X^b}, \kappa^Y\}. \tag{12}$$

In non-object areas, the original distribution of diffusion fusion should be maintained:

$$\{Z_t^b\}^{re-modu} = (1 - M) \cdot Z_t^b + M \cdot \{Z_t^b\}^{mod}. \tag{13}$$

The modulated fused image enhances the saliency of objects compared to the before, making it more suitable for subsequent advanced tasks. We prove this in the re-modulation verification section.

## 4 Experiments

**Configuration.** We evaluate our method on two typical multi-modal image fusion scenarios: infrared and visible image fusion (IVIF) and medical image fusion (MIF). For IVIF, we use the MSRS dataset [43], with $485$ training and $100$ testing image pairs. For MIF, we use the *Harvard medicine dataset*[3] with 160 training and 50 testing image pairs, covering CT-MRI, PET-MRI, and SPECT-MRI. Data augmentation like random flipping and cropping increases the training pairs to $12,888$ for IVIF and $6,408$ for MIF. Besides, generalization is evaluated on 60 pairs from LLVIP [15] and 25 pairs from RoadScene [49]. Competitors include 9 methods: RFN-Nest [20], GANMcC [31], SDNet [53], U2Fusion [49], TarDAL [25], DeFusion [23], LRRNet [21], DDFM [63], and MRFS [58]. Five metrics are used: EN [39], AG [3], SD [38], SCD [2], and VIF [11]. The Adam optimizer with a learning rate of $2e^{-5}$ is used for parameter updates. Experiments are conducted on an NVIDIA RTX 3090 GPU and a 3.80 GHz Intel i7-10700K CPU.

---

[3] https://www.med.harvard.edu/AANLIB/home.html

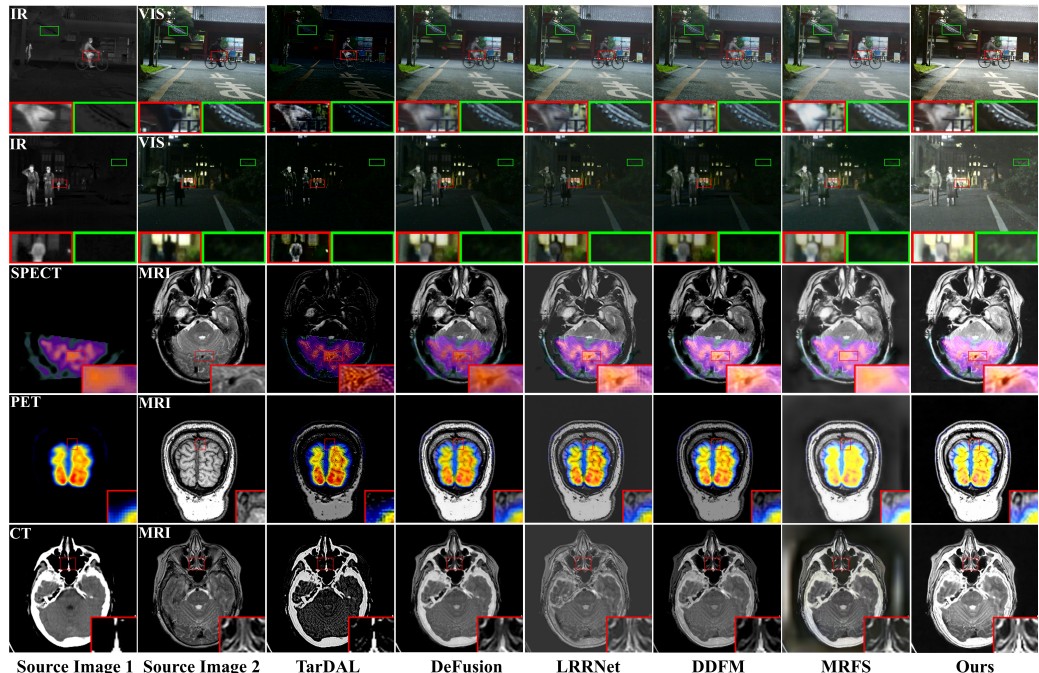

Figure 3: Visual comparison of image fusion methods.

Table 1: Quantitative comparison of image fusion methods. **Blod**: the best; underline: second best.

| Methods | MSRS DataSet | | | | | Havard Medicine Dataset | | | | |
|---|---|---|---|---|---|---|---|---|---|---|
| | EN ↑ | AG ↑ | SD ↑ | SCD ↑ | VIF ↑ | EN ↑ | AG ↑ | SD ↑ | SCD ↑ | VIF ↑ |
| RFN-Nest (InF'21) | 5.89 | 1.84 | 26.03 | 1.41 | 0.63 | 5.34 | 4.07 | 63.65 | 1.58 | 0.43 |
| GANMcC (TIM'21) | 6.03 | 1.91 | 25.58 | 1.37 | 0.65 | 5.38 | 5.13 | 55.00 | 1.11 | 0.43 |
| SDNet (IJCV'21) | 4.90 | 2.33 | 16.35 | 0.91 | 0.49 | 5.56 | 6.22 | 46.64 | 0.48 | 0.38 |
| U2Fusion (TPAMI'22) | 5.19 | 2.46 | 24.82 | 1.21 | 0.52 | 5.22 | 6.08 | 53.20 | 1.03 | 0.41 |
| TarDAL (CVPR'22) | 3.30 | 2.04 | 18.52 | 0.63 | 0.15 | 5.66 | 5.13 | 41.94 | 1.12 | 0.18 |
| DeFusion (ECCV'22) | 6.22 | 2.31 | 32.34 | 1.36 | 0.75 | 4.96 | 4.47 | 55.45 | 0.93 | 0.48 |
| LRRNet (TPAMI'23) | 5.89 | 2.19 | 26.64 | 0.75 | 0.52 | 5.34 | 5.39 | 45.89 | 0.59 | 0.40 |
| DDFM (ICCV'23) | 5.81 | 2.65 | 24.98 | 1.37 | 0.62 | 5.00 | 5.04 | 63.53 | 1.59 | 0.48 |
| MRFS (CVPR'24) | 6.91 | 2.67 | 40.95 | 1.23 | 0.75 | **7.24** | 4.41 | 70.75 | 1.53 | 0.41 |
| Ours (Text-DiFuse) | **7.08** | **3.31** | **47.44** | **1.44** | **0.76** | 6.44 | **7.31** | **80.19** | **1.69** | **0.49** |

**Comparative Experiments.** We first compare the basic version of our Text-DiFuse with current state-of-the-art fusion methods, and the qualitative results are shown in Fig. 3. The first two rows depict IVIF results, demonstrating our method's ability to correct color casts, restore scene information under low-light conditions, and suppress noise. The last three rows display MIF results, which show that our method can highlight physiological structure information while maintaining functional distribution. In contrast, competitors are unable to achieve such information recovery and still suffer significantly weakened appearance. The quantitative results in Table 1 demonstrate our method's advantages over other fusion techniques. For further fairness, we introduce state-of-the-art low-light enhancement (CLIP-LIT [24]), denoising (SDAP [36]), and white balance (AWB [1]) algorithms as the pre-processing steps for these competitors, with results presented in Fig. 4 and Table 2. Clearly, our method still outperforms these comparative methods. This is because these added pre-processing steps for information recovery are entirely independent of information fusion, so they cannot mine habits that are more conducive to modal complementarity, leading to limited performance.

**Generalization Evaluation.** Next, we directly test the model trained on the MSRS dataset on the LLVIP and RoadScene datasets to evaluate the generalization ability of the proposed method. We select a daytime scene with overexposure and a low-light nighttime scene, and the qualitative results are shown in Fig. 5. It can be observed that our Text-DiFuse still maintains high-quality degradation removal and information fusion capabilities. In particular, it has two-way information recovery functions such as overexposure correction and low-light enhancement, producing visually satisfying

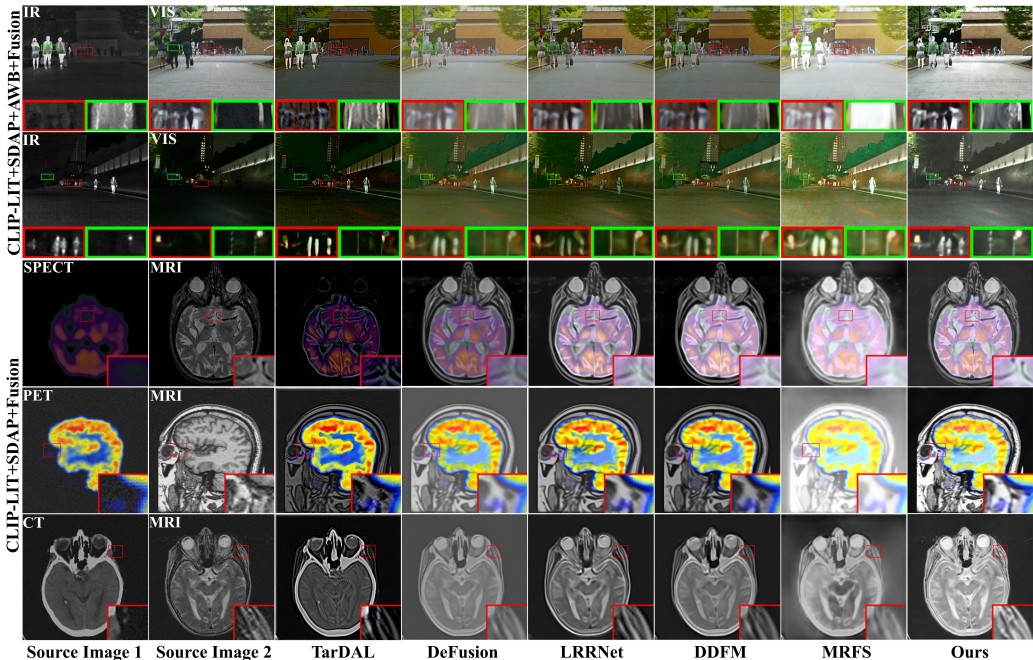

Figure 4: Visual comparison of enhancement plus image fusion methods.

Table 2: Quantitative comparison of enhancement plus image fusion methods.

| Methods | | MSRS Dataset | | | | | Havard Medicine Dataset | | | | |
|---|---|---|---|---|---|---|---|---|---|---|---|
| | | EN ↑ | AG ↑ | SD ↑ | SCD ↑ | VIF ↑ | EN ↑ | AG ↑ | SD ↑ | SCD ↑ | VIF ↑ |
| | RFN-Nest | 6.43 | 2.23 | 27.17 | 1.38 | 0.60 | 5.72 | 4.11 | 77.46 | 1.64 | 0.35 |
| | GANMcC | 6.25 | 2.06 | 24.55 | 1.31 | 0.57 | 5.80 | 5.28 | 66.37 | 1.19 | 0.31 |
| | SDNet | 5.84 | 2.99 | 20.26 | 1.08 | 0.52 | 5.91 | 6.00 | 60.83 | 1.15 | 0.30 |
| CLIP-LIT | U2Fusion | 6.55 | 3.55 | 29.08 | 1.32 | 0.58 | 5.68 | 6.09 | 71.59 | 1.56 | 0.32 |
| SDAP | TarDAL | 5.29 | 4.42 | 25.22 | 1.00 | 0.35 | 6.11 | 4.81 | 36.54 | 0.69 | 0.23 |
| AWB | DeFusion | 6.31 | 2.07 | 25.52 | 1.16 | 0.59 | 6.08 | 4.27 | 67.77 | 1.38 | 0.35 |
| | LRRNet | 6.55 | 2.68 | 31.19 | 1.13 | 0.54 | 5.86 | 5.23 | 62.91 | 1.34 | 0.21 |
| | DDFM | 6.39 | 2.43 | 26.40 | 1.16 | 0.60 | 5.70 | 4.48 | 77.40 | 1.64 | 0.35 |
| | MRFS | 6.84 | 2.86 | 32.28 | 1.28 | 0.58 | 7.18 | 4.19 | 87.53 | 1.50 | 0.31 |
| Ours (Text-DiFuse) | | 7.08 | 3.31 | 47.44 | 1.44 | 0.76 | 6.44 | 7.13 | 80.19 | 1.69 | 0.49 |

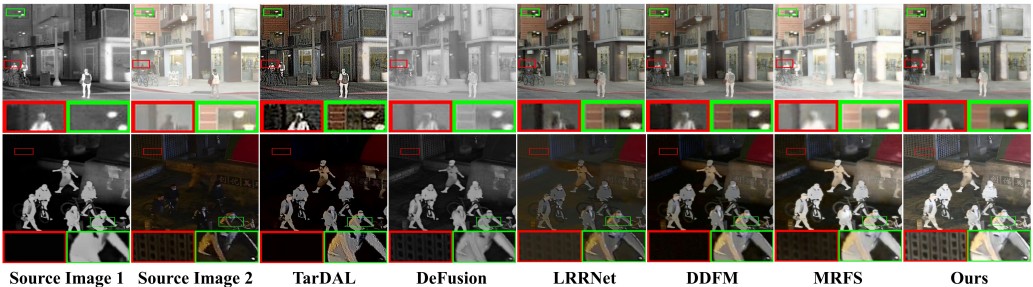

Figure 5: Visual results of generalization evaluation.

fused results. By comparison, other competitors lose useful information masked by overexposure or low light. We further prove the good generalization ability of our method in Table 3. In general, these results indicate that our Text-DiFuse can be applied more reliably in real scenarios.

**Re-modulation Verification.** We verify the performance gains brought by our text-controlled fusion re-modulation strategy on the MFNet dataset [10]. We select 6 state-of-the-art RGB-T segmentation methods for comparison, *i.e.*, MFNet [10], FEANet [4], EGFNet [7], CMX [59], GMNet [65], and MDRNet [48]. Besides, we train SegNext [9] on infrared images, visible images, the fused

Table 3: Quantitative comparison of generalization ability.

| Methods | LLVIP Dataset | | | | | RoadScene Dataset | | | | |
|---|---|---|---|---|---|---|---|---|---|---|
| | EN ↑ | AG ↑ | SD ↑ | SCD ↑ | VIF ↑ | EN ↑ | AG ↑ | SD ↑ | SCD ↑ | VIF ↑ |
| RFN-Nest | 6.37 | 2.24 | 26.66 | 1.63 | 0.73 | 7.37 | 2.62 | 46.77 | 1.66 | 0.58 |
| GANMcC | 6.24 | 2.09 | 27.02 | 1.59 | 0.65 | 7.24 | 3.58 | 43.68 | 1.39 | 0.57 |
| SDNet | 6.00 | 2.74 | 23.05 | 1.24 | 0.62 | 7.18 | 4.86 | 40.63 | 1.16 | 0.66 |
| U2Fusion | 5.52 | 2.69 | 21.12 | 1.32 | 0.61 | 7.32 | 4.92 | 43.99 | 1.49 | 0.66 |
| TarDAL | 3.85 | 2.59 | 23.05 | 0.92 | 0.22 | 7.35 | 11.84 | 52.30 | 0.97 | 0.47 |
| DeFusion | 6.46 | 2.36 | 29.48 | 1.48 | 0.82 | 6.97 | 2.85 | 35.96 | 0.98 | 0.59 |
| LRRNet | 5.67 | 2.28 | 19.49 | 1.06 | 0.57 | 7.19 | 3.55 | 44.01 | 1.47 | 0.58 |
| DDFM | 6.46 | 3.51 | 30.64 | 1.72 | 0.70 | 7.30 | 3.63 | 44.19 | 1.57 | 0.65 |
| MRFS | 7.00 | 2.34 | 40.90 | 1.67 | 0.86 | 7.18 | 2.70 | 46.57 | 1.20 | 0.52 |
| Ours (Text-DiFuse) | 7.08 | 3.99 | 41.78 | 1.73 | 0.87 | 7.46 | 2.96 | 52.84 | 1.67 | 0.66 |

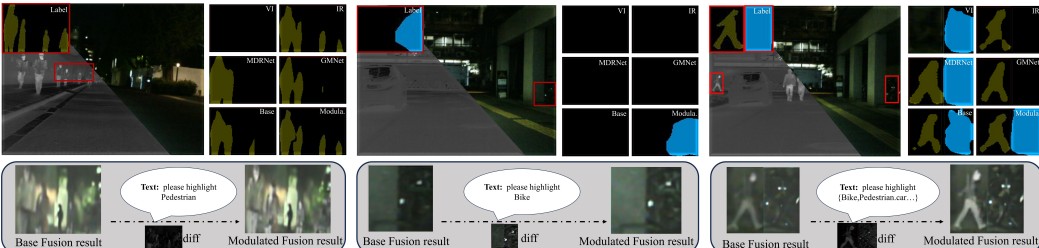

Figure 6: Visual results of re-modulation verification.

Table 4: Quantitative verification of re-modulation on semantic segmentation.

| Segmentation | Source | Background | Car | Person | Bike | Curve | Car Stop | Cuardrail | Color cone | Bump | mIoU |
|---|---|---|---|---|---|---|---|---|---|---|---|
| MFNet | RGB-T | 96.26 | 60.95 | 53.44 | 43.14 | 22.94 | 9.44 | 0.00 | 18.80 | 23.47 | 36.49 |
| FEANet | RGB-T | 98.00 | 87.41 | 70.30 | 62.74 | 45.33 | 29.80 | 0.00 | 29.07 | 48.95 | 55.28 |
| EGFNet | RGB-T | 98.01 | 87.84 | 71.12 | 61.08 | 46.48 | 22.10 | 6.64 | 55.35 | 47.12 | 54.76 |
| CMX-B2 | RGB-T | 97.39 | 84.23 | 67.12 | 56.93 | 41.11 | 39.56 | 18.94 | 48.84 | 54.42 | 58.31 |
| GMNet | RGB-T | 98.00 | 86.46 | 73.05 | 61.72 | 43.96 | 42.25 | 14.52 | 48.70 | 47.72 | 57.34 |
| MDRNet | RGB-T | 97.90 | 87.07 | 69.81 | 60.87 | 47.80 | 34.18 | 8.21 | 50.18 | 54.98 | 56.78 |
| SegNext-Base | IR | 97.79 | 84.89 | 70.73 | 56.29 | 41.94 | 24.15 | 7.60 | 35.91 | 48.64 | 51.99 |
| | VI | 97.93 | 88.29 | 62.42 | 63.67 | 35.34 | 36.95 | 5.77 | 51.20 | 47.74 | 54.37 |
| | Our basis | 98.11 | 88.66 | 70.00 | 64.30 | 43.07 | 30.25 | 11.95 | 55.14 | 56.27 | 57.53 |
| | Our modulatable | 98.18 | 88.32 | 72.23 | 65.02 | 44.79 | 33.11 | 13.76 | 56.32 | 55.97 | 58.63 |

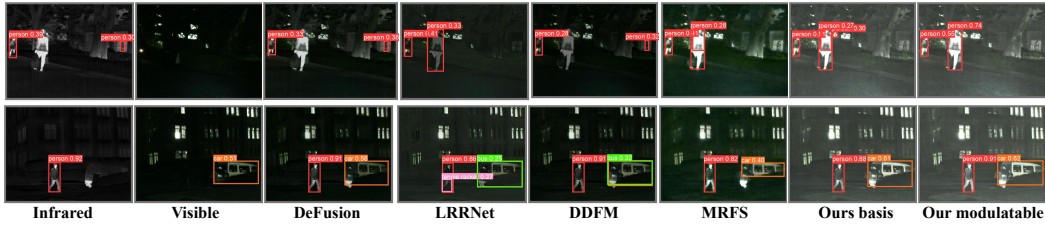

Figure 7: Visual verification in detection scenario.

images generated by the basic version of our method, and the modulatable version respectively to achieve segmentation. It can be seen from Fig. 6 that different language commands derive customized fused results, which promote the completeness and accuracy of semantic segmentation while visually highlighting the objects of interest. The quantitative results in Table 4 further prove that our re-modulation strategy can improve the semantic attributes, achieving the best segmentation scores.

**Semantic Verification on Detection.** We further verify the semantic gain brought by text modulation on the object detection task. Specifically, the MSRS dataset [43] is used, which includes pairs of infrared and visible images with two types of detection labels: person and car. Therefore, the text instruction is formulated as: "Please highlight the person and car", which guides our method to enhance the representation of these two types of objects in the fused image. Then, we adopt the YOLO-v5 detector to perform object detection on infrared images, visible images, and fused images generated by various image fusion methods. The visual results are presented in Fig. 7, in which more complete cars and people can be detected from our fused images while showing higher class

Table 5: Quantitative verification in detection scenario.

| Detection | IR | VIS | DeFusion | LRRNet | DDFM | MRFS | Our basis | Our modulatable |
|---|---|---|---|---|---|---|---|---|
| mAP@0.5 | 71.9 | 74.8 | 86.6 | 86.3 | 88.6 | 82.0 | 87.3 | **89.7** |
| mAP@[0.5:0.95] | 48.4 | 47.3 | 60.1 | 58.9 | 59.4 | 53.2 | 56.3 | **60.9** |

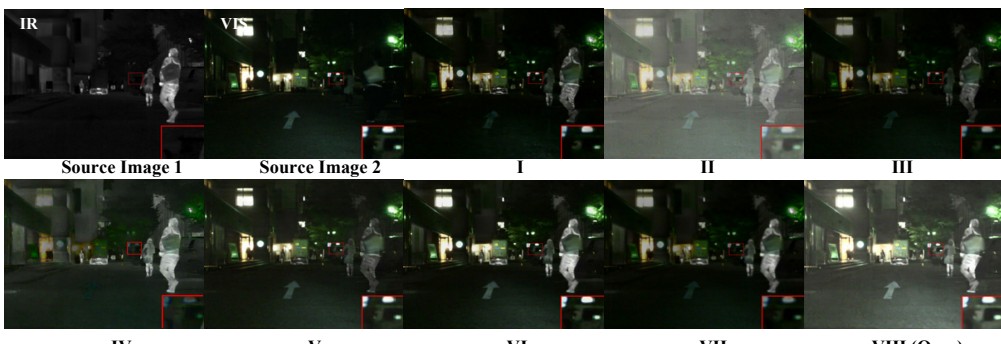

Figure 8: Visual results of ablation studies.

Table 6: Quantitative results of ablation studies.

| Index | Diff. | $\mathcal{L}_{int}$ | $\mathcal{L}_{grad}$ | FCM | EN | AG↑ | SD↑ | SCD↑ | VIF↑ |
|---|---|---|---|---|---|---|---|---|---|
| I | ✓ | ✓ | ✓ | ✗/max | 5.71 | 1.90 | 25.66 | 1.30 | 0.49 |
| II | ✓ | ✓ | ✓ | ✗/add | 6.08 | 1.99 | 20.14 | 1.11 | 0.58 |
| III | ✓ | ✓ | ✓ | ✗/mean | 5.60 | 1.49 | 20.71 | 1.15 | 0.56 |
| IV | ✓ | ✓ | ✓ | ✗/variance | 5.91 | 1.90 | 27.59 | 0.91 | 0.46 |
| V | ✓ | ✓ | ✗ | ✓ | 6.20 | 2.75 | 33.78 | 1.42 | 0.73 |
| VI | ✓ | ✗ | ✓ | ✓ | 6.67 | 3.25 | 45.60 | 1.43 | 0.76 |
| VII | ✗/AE | ✓ | ✓ | ✓ | 6.37 | 2.26 | 37.48 | 1.42 | 0.69 |
| VIII | ✓ | ✓ | ✓ | ✓ | **7.08** | **3.31** | **47.44** | **1.44** | **0.76** |

confidence. Furthermore, we provide quantitative detection results in Table 5. It can be seen that the highest average accuracy is obtained from our fused images, demonstrating the benefits of text modulation. Overall, these results indicate that text control indeed provides significant semantic gains, benefiting downstream tasks.

**Ablation Studies.** We conduct ablation studies to verify the effectiveness of specific designs, involving eight variants: **I**: removing FCM with using maximum rule; **II**: removing FCM with using addition rule; **III**: removing FCM with using mean rule; **IV**: removing FCM with using variance-based rule [34]; **V**: removing $\mathcal{L}_{grad}$; **VI**: removing $\mathcal{L}_{int}$; **VII**: removing diffusion with using AE route; **VIII**: our full model. The visual results in Fig. 7 show that removing any of these designs results in a reduction of visual satisfaction. The quantitative scores in Table 5 also support this view. Overall, these designs in our Text-DiFuse collectively guarantee advanced fusion performance.

## 5 Conclusion

This paper proposes a new interactive multi-modal image fusion framework based on the text-modulated diffusion model. On the one hand, it is the first to develop an explicit coupling paradigm for information fusion and diffusion models, achieving the integration of multi-modal beneficial information while removing composite degradation. On the other hand, a text-controlled fusion re-modulation strategy is designed. It incorporates text combined with the zero-shot location module into the diffusion fusion process, supporting users' language control to enhance the perception of objects of interest. Extensive experiments demonstrate that our method achieves better performance than current methods, effectively improving the visual quality and semantic attributes of fused results.

## 6 Acknowledgement

This work was supported by NSFC (62276192).

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
