# OpenReview forum: "Text-DiFuse: An Interactive Multi-Modal Image Fusion Framework based on Text-modulated Diffusion Model"
_NeurIPS.cc/2024/Conference — NeurIPS 2024 spotlight_

### Official Review · Reviewer_NcHo · 2024-07-03

**Soundness:** 4
**Presentation:** 4
**Contribution:** 4
**Rating:** 7
**Confidence:** 5

**Summary:**

A new paradigm of multi-modal image fusion named Text-DiFuse is introduced, based on the diffusion model. The paradigm embeds a mechanism for aggregating feature-level multi-modal image information into the diffusion process of degrading multi-modal images, addressing the optimization gap between "degradation removal" and "multi-modal information fusion". Additionally, a zero-shot model is introduced to modulate the fusion strategy based on user-input target text, enhancing the saliency of the target of interest. The conducted experiments suggest significant improvements in both human visual perception and advanced computer vision tasks.

**Strengths:**

1)	Embedding the mechanism of aggregating feature-level information into multiple diffusion processes to fuse multi-modal information is interesting. It is foreseeable that this diffusion paradigm produces fused images with better fidelity compared to methods based on likelihood-constrained diffusion models.
2)	The coupled approach effectively resolves the issue of compound degradation in the process of multi-modal fusion, as evidenced by experimental results demonstrating significant advantages over the sequential approach.
3)	The authors emphasize the importance of foreground targets in advanced visual tasks and propose enhancing target saliency through zero-shot assisted re-modulation. This approach diverges from traditional uniform fusion rules, demonstrating effectiveness.
4)	This approach shows strong applicability. It demonstrates superior performance in multiple tasks including infrared and visible image fusion, medical image fusion, and polarization image fusion.

**Weaknesses:**

1)	After the diffusion model is effectively trained, the sampling process can follow different step intervals. The information fusion in this method is integrated into the diffusion process, but the article does not seem to specify the sampling interval at which the results are obtained. Also, this article does not discuss the impact of the sampling interval on the fusion performance.
2)	The presentation is slightly unclear. For example, from Equation 2 to Equation 6, both the features and the images carry the condition N that represents the degradation. Why does equation 7 no longer include N? Why can it be considered that the degradation has been removed at this point?
3)	In Table 2 and Figure 4, some existing image restoration methods are cascaded in front of the fusion method to promote fairness in comparison, such as low-light enhancement (CLIP-LIT), denoising (SDAP), and white balance (AWB) algorithms.
Please explain the choice of the order in which they are connected in series, i.e. why low light enhancement first, then denoising, and finally white balance.
4)	Modulating the salience of targets of interest in the fusion process through language is novel. Intuitively, I think the improvement in semantic properties brought about by this modulation is widespread. Currently, the effectiveness of language modulation has only been verified in the semantic segmentation scenario. It is recommended to provide an evaluation in the object detection scenario to further verify its role.

**Questions:**

Please refer to the weaknesses part.

**Limitations:**

The authors have analyzed the limitations and potential negative impact.

---

> ### Author Rebuttal · Authors · 2024-08-06
>
> Q1: Sampling interval and its impact.\
> Reply: In our method, image restoration and information integration are mutually coupled. This is reflected in the physical connection, where a fusion control module is embedded within the internal structure of the diffusion model. Once all the networks are trained, we can follow the standard diffusion model testing procedure, which involves performing T steps of continuous sampling. It is worth noting that information fusion needs to be performed at each sampling step. In this case, the only factor affecting the final fusion result is the number of sampling steps. More sampling steps mean better performance, but they also result in significant time consumption. Therefore, setting an appropriate number of sampling steps is a matter worth discussing.\
> In tasks where the ground truth is available, the number of sampling steps can be well determined by checking whether the generated results are sufficiently close to the ground truth. However, for the image fusion task, where ground-truth data do not exist, we rely on visual perception and multiple no-reference metrics to make the assessment. Specifically, we set the number of sampling steps to 2, 3, 4, 5, 10, 25, 500, and 1000, with qualitative and quantitative results shown in Figs. r7 and r8. Notably, each metric is normalized along the step dimension for easier presentation. It can be observed that as the number of steps increases, noise is gradually removed and the scene texture becomes increasingly refined. Corresponding to the quantitative results, 25 steps achieve good performance saturation, with subsequent increases in the number of steps resulting in only slight fluctuations in scores. Note that the only exception is AG, as it is affected by noise during the diffusion process. Therefore, in our experimental section, the number of sampling steps is set to 25.
>
> Q2: Removal of degradation symbols $N$ in Eq. (7).\
> Reply: There are two conceptual differences that need to be clarified first. In all the equations, the symbol $N$ refers to the degradation from the source images, specifically including improper lighting, color distortion, and random noise. In contrast, the noise in the intermediate results obtained from continuous sampling of the diffusion model arises from the Gaussian noise assumption of the diffusion theory itself. Eqs. (2)-(6) represent the process of encoding, fusion, decoding, and the estimation of mean and variance for degraded multi-modal images. In this process, the objects being processed contain the degradation from source images, so they all include the condition $N$. Differently, Eq. (7) represents the intermediate result obtained after a single complete sampling step. Therefore, even though early sampling steps still contain Gaussian noise following the diffusion assumption (see Fig. r7), it does not need to include the symbol $N$. We will clarify their distinctions in the final version to avoid misunderstandings.
>
> Q3: Concatenate order of image restoration algorithm.\
> Reply: In Table 2, we introduce three image restoration algorithms as preprocessing steps of other comparative image fusion methods, including CLIP-LIT, SDAP, and AWB. Among them, CLIP-LIT is a low-light enhancement algorithm, SDAP is a denoising algorithm, and AWB is a white balance algorithm. In the experiment, we follow the processing sequence of low-light enhancement first, followed by denoising, and finally white balance. The choice of this sequence is related to the dependencies among the three types of degradations we are focusing on. Specifically, in low-light images, both scene content and degradation present low-intensity properties, and the signal-to-noise ratio is low. The deep entanglement of noise, color distortion, and useful signals makes degradation removal more challenging. Thus, we first use CLIP-LIT to improve exposure, thereby reducing the difficulty of denoising and color correction. Furthermore, color correction based on white balance requires locating the white light source, and noise interferes with the accuracy of finding the light source. Therefore, we then perform SDAP to remove noise. After addressing exposure and noise, AWB is applied last to achieve color correction.
>
> Q4: Verification of text control on the object detection.\
> Reply: Our method supports text control, enabling the enhancement of the salience of objects of interest based on instructions. Following the reviewer's suggestion, we further verify the semantic gain brought by text modulation on the object detection task. Specifically, the MSRS dataset [r1] is used, which includes pairs of infrared and visible images with two types of detection labels: person and car. Therefore, the text instruction is formulated as: "Please highlight the person and car," which guides our method to enhance the representation of these two types of objects in the fused image. Then, we adopt the YOLO-v5 detector to perform object detection on infrared images, visible images, and fused images generated by various image fusion methods. The visual results are presented in Fig. r6, in which more complete cars and people can be detected from our fused images while showing higher class confidence. Furthermore, we provide quantitative detection results in Table r5. It can be seen that the highest average accuracy is obtained from our fused images, demonstrating the benefits of text modulation. Overall, these results indicate that text control indeed provides significant semantic gains, benefiting downstream tasks.\
> [r1] PIAFusion: A progressive infrared and visible image fusion network based on illumination aware. Information Fusion, 2022.

---

> > ### Comment · Reviewer_NcHo · 2024-08-12
> >
> > The response is clear and my concerns are addressed. In particular, the effectiveness of semantic attribute improvement is verified in the object detection scenario. I think this observation is inspiring. I also see that the authors perform an additional comparison by using an all-in-one image enhancement algorithm InstructIR, still showing this work's effectiveness. Consequently, I plan to keep my original rating and recommend accepting this paper.

---

> > > ### Author Response · Authors · 2024-08-13
> > >
> > > Thank you very much for your positive feedback on our paper. The textual modulation enhances generalized semantic attributes, illustrating that abstract text encompasses rich semantic information and can aid in improving machine vision perception. This observation can inspire the design of methods for various high-level visual tasks. In the future, we will explore controllable semantic decision paradigms based on text integration, achieving various interesting functions, such as text-assisted, text-specified, and text-deception decisions. Furthermore, implicit integration of information restoration and fusion is indeed more attractive than explicit concatenation. This is not only because of its advanced performance but also due to its ability to handle multiple tasks with a single set of parameters. We commit that if this paper is accepted, all clarifications provided in the rebuttal will be incorporated into the camera-ready version.

---

### Official Review · Reviewer_RoRE · 2024-07-08

**Soundness:** 3
**Presentation:** 2
**Contribution:** 3
**Rating:** 7
**Confidence:** 5

**Summary:**

This work focuses on the topic of multi-modal image fusion. Two innovations enhance the performance of the fusion. One is the clever integration of information fusion into the diffusion process. This coupling way enables the fusion function to resist degradation. The other is the introduction of a text-based fusion remodulation strategy. This changes the limitation of previous fusion methods that could only use fixed mappings, allowing for the dynamic adjustment of the fused image based on specific requirements. This remodulation also enhances semantic attributes, improving the scores of the semantic segmentation task.

**Strengths:**

1. Integrating information fusion into the diffusion process is novel. Especially, each sampling step triggers an information fusion, which enhances the sufficiency of information fusion. This coupling can ensure the robustness of information fusion, addressing challenges such as low light, noise, and color cast.
2. The introduction of multi-modal large models is interesting, particularly the ability to remodulate fused images using textual commands. This capability could potentially facilitate the flexible deployment of the proposed method across different application requirements. The demonstration of enhanced semantic attributes and improved semantic segmentation performance is good.
3. Overall, the experiments are relatively sufficient. The comparative experiments include both baseline comparisons and pre-enhancement comparisons, which are important for ensuring fairness.
4. The code is provided, which helps in reproducing the performance.

**Weaknesses:**

1. On page 5, line 174, the source data used for fusion contains degradation, [{Xb,Y}|N]. My question is, in Equations (9) and (10), where do the clean {Xb,Y} used to guide the fusion come from? Is there a multi-modal dataset that contains paired degraded and clean data? The paper seems to lack an explanation for this.
2. The forward process of the diffusion model involves T steps of noise addition, while the reverse process consists of T steps of iterative sampling. Is the Z0 obtained in equation (8) a hypothetical Z0 derived from the diffusion relation at each sampling, or is it the Z0 after completing the full T steps of sampling? This determines the object of the constraints in the loss functions (9) and (10). It would be better to provide a detailed discussion on this.
3. Only after the T steps of sampling can the data without degradation be obtained. So why can Z_{t-1}^b in equation (7) be considered free from degradation N?
4. It's understandable that using textual modulation to control the desired targets of interest can enhance semantic attributes. My question is whether these enhanced semantic attributes can be generalized. In other words, can it also be effective in other high-level visual tasks besides semantic segmentation?
5. Typo: The Zt on the left side of equation (8) seems to have a missing superscript b.

**Questions:**

Please answer the question raised in Weaknesses.

**Limitations:**

Yes, there are discussions about the limitations and potential negative societal impacts.

---

> ### Author Rebuttal · Authors · 2024-08-06
>
> Q1: Clean data for the loss construction.\
> Reply: Constructing Eqs. (9) and (10) actually involves very stringent data requirements. Specifically, they require a pair of degraded multi-modal images describing the same scene, along with their corresponding clean versions. Unfortunately, such a dataset is currently not available. To alleviate this challenge, we adopt a two-step strategy. Specifically, we first pre-train the diffusion model to learn the image restoration capability. In this step, we only need degraded-clean image pairs, without the need for paired multi-modal images that describes the same scene. Once the diffusion model is trained, it can be used to process existing degraded multi-modal image fusion datasets, to generate the required clean multi-modal image pairs. At this point, all the data required for constructing Eqs. (9) and (10) has been obtained.
>
> Q2: Source of the constrained fused image in Eq. (8).\
> Reply: Our method couples image restoration and information integration by inserting a fusion control module within the diffusion model. In other words, each step of sampling will be accompanied by an information fusion. Theoretically, the final fused image requires T steps of continuous iterative sampling to obtain. However, during training, it is inefficient to wait for T steps of sampling to obtain the result and then construct the loss functions of Eqs. (9) and (10). Therefore, we customize Eq. (8) based on the diffusion relationship. According to it, we can derive the corresponding fake final fused image from the results of any step of sampling and apply the corresponding fusion constraints.
>
> Q3: Omission of degradation $N$ in Eq. (7).\
> Reply: Indeed, diffusion models often require a certain number of sampling steps to progressively remove the noise. However, it is needed to clarify that the degradation symbol $N$ in Eqs. (2)-(6) is not the same thing as the noise in the diffusion process. Specifically, the symbol $N$ refers to the degradation from the source images, specifically including improper lighting, color distortion, and random noise. They are used as the conditions of the diffusion model, included in the source image and fed into the denoising network. In contrast, the noise in the intermediate results obtained from continuous sampling of the diffusion model arises from the Gaussian noise assumption of the diffusion theory itself. Eqs. (2)-(6) represent the process of encoding, fusion, decoding, and the estimation of mean and variance for degraded multi-modal images. In this process, the objects being processed contain the degradation from source images, so they all include the condition $N$. Differently, Eq. (7) represents the intermediate result obtained after a single complete sampling step. Therefore, even though early sampling steps still contain Gaussian noise following the diffusion assumption (see Fig. r7), it does not need to include the symbol $N$. We will clarify their distinctions in the final version to avoid misunderstandings.
>
> Q4: Generalization of semantic attributes.\
> Reply: Our method supports the use of textual instructions to re-modulate the information fusion process. Its purpose is to increase the salience of the object of interest, and enhance its presentation quality on the fused image. In our method, no specific downstream task is used to guide this remodulation process, so the gain in semantic attributes is not fixed to any particular task. In other words, beyond the semantic segmentation validation presented in the main text, the semantic attribute gain achieved through textual remodulation can certainly be generalized to other downstream tasks.\
> For proving this point, we implement application experiments on the object detection task. Specifically, the MSRS dataset [r1] is used, which includes pairs of infrared and visible images with two types of detection labels: person and car. Therefore, the text instruction is formulated as: "Please highlight the person and car," which guides our method to enhance the representation of these two types of objects in the fused image. Then, we adopt the YOLO-v5 detector to perform object detection on infrared images, visible images, and fused images generated by various image fusion methods. The visual results are presented in Fig. r6, in which more complete cars and people can be detected from our fused images while showing higher class confidence. Furthermore, we provide quantitative detection results in Table r5. It can be seen that the highest average accuracy is obtained from our fused images, demonstrating the benefits of text modulation. Overall, these results indicate that text control indeed provides significant semantic gains, benefiting downstream tasks.\
> [r1] PIAFusion: A progressive infrared and visible image fusion network based on illumination aware. Information Fusion, 2022.
>
> Q5: Typos.\
> Reply: Thanks for pointing out these issues. We will carefully correct all typos in the final version and further enhance the presentation of the figures and tables.

---

> > ### Comment · Reviewer_RoRE · 2024-08-12
> > **To author's response**
> >
> > Thanks for the effort to clarify my questions in the provided rebuttal. Using the two-step strategy to address the limitation of data unavailability is clever, and I'm pleased to see the general semantic attributes brought by textual modulation. Therefore, I'm inclined to accept this paper. In the camera-ready version, please include the provided clarifications about the source of Z0 in loss functions (9) and (10) and the degradation N.

---

> > > ### Author Response · Authors · 2024-08-13
> > >
> > > We truly appreciate your efforts in improving our paper. If this paper is accepted, we plan to incorporate the following revisions into the camera-ready version in response to your recommendations:
> > >
> > > 1. Explain the source of clean data for constructing loss functions: We adopt a two-step strategy to alleviate this challenge of unavailable data. The core idea of the two-step strategy is to relax the high data requirements by pre-training a generative model with limited available data. Then, this pre-trained model allows for the production of data that is not available in reality.
> > >
> > > 2. Explain the source of $Z_0$: We customize Eq. (8) based on the diffusion relationship, so we can derive the corresponding fake final fused image $Z_0$ from the results of any sampling step and apply the corresponding fusion constraints. We also consider using the results after all sampling as Z0 in loss functions (9) and (10). The quantitative results are reported below, demonstrating the advantages of our method compared to the full sampling strategy.
> > > | $Z_0$ | EN | AG| SD| SCD| VIF|
> > > | ---- | ---- | ---- | ---- | ---- | ---- |
> > > | Full Sampling  | 5.93 | 1.74 | 23.99 | 1.26 | 0.63 |
> > > | Ours | **7.08** | **3.31** | **47.44** | **1.44** | **0.76** |
> > >
> > > 3. Explain the degradation $N$: The degradation symbol in Eqs. (2)-(6) is not the same thing as the noise in the diffusion process. The former refers to the degradation from the source images, specifically including improper lighting, color distortion, and random noise. In contrast, the latter comes from the Gaussian noise assumption of diffusion theory itself.
> > >
> > > 4. Verify textual modulation in object detection: We implement application experiments on the object detection task to verify the generalization of semantic attributes. Experimental results indicate that textual modulation indeed enhances semantic information, thereby benefiting downstream tasks.  In the future, we will explore controllable semantic decision paradigms based on text integration, achieving various interesting functions, such as text-assisted, text-specified, and text-deception decisions.
> > >
> > > We greatly appreciate your positive feedback on our work. If you have any further questions or concerns, please feel free to contact us.

---

### Official Review · Reviewer_TtBn · 2024-07-09

**Soundness:** 4
**Presentation:** 3
**Contribution:** 4
**Rating:** 6
**Confidence:** 5

**Summary:**

This paper addresses two primary challenges in multimodal image fusion: the mixed degradation of modalities and the insufficient salience of target objects. It proposes two methods to tackle these challenges: feature-level fusion diffusion and the re-modulation of fusion rules in target areas using a zero-shot segmentation model. They implement adequate experiments for evaluation, and the results demonstrate this method's advanced performance across various aspects, including the visual and semantic.

**Strengths:**

+ The mixed degradation of modalities and the insufficient salience of target objects are two interesting problems in multimodal image fusion. This paper’s discussion and solution of these two problems may promote the usability of fusion methods in real scenarios.
+ The information fusion at the feature level is integrated into the diffusion process, which effectively realizes the degradation removal.
+ The customized object highlighting strategy based on the zero-shot segmentation model is flexible. In particular, its gain in semantic attributes will increase the usability of the fused image in downstream tasks.
+ This paper conducts lots of comparative experiments and ablation studies on the overall method.
+ The narrative of this paper is comprehensive and clear. For me, it's easy to follow.

**Weaknesses:**

- This paper mentioned that the diffusion model is pre-trained to enable the denoising network to have the degradation removal function. However, details about the construction of the data used to train the diffusion model are missing. They need to describe this process to make the overall approach clearer.
- This paper focuses on multimodal image fusion, being reflected in the title. In the main text, the proposed method is evaluated in two scenarios: infrared and visible image fusion and medical image fusion. In the supplementary materials, they further provide experiments on polarization image fusion. I am curious whether the applicable scenarios of the proposed method can be further expanded, such as the typical fusion of near-infrared and visible bands.
- The experiments on polarization image fusion only provide visual results, and it would be better to add a quantitative evaluation.
- I noticed that the proposed method separates the chrominance component and the brightness component, and then performs de-degradation on them separately. An explanation of why this operation is needed should be given. Perhaps an ablation experiment could more intuitively show the effect of this operation.
- There are some minor typos, such as potential misspellings of dataset names in Tables 1 and 2. In addition, there seems to be a lack of underline on AG's second place.

**Questions:**

1. How were the degradation condition data constructed, were paired supervised datasets used or synthetic datasets?
2. Has there been an attempt to evaluate the fusion on the fusion of near-infrared and visible bands?
3. Could you provide the quantitative results of polarization image fusion?
4. The separation of chrominance and brightness requires more explanation.

**Limitations:**

Limitations and broader impacts have been included.

---

> ### Author Rebuttal · Authors · 2024-08-06
>
> Q1: Dataset for training diffusion model.\
> Reply: In our work, acquiring image restoration capability depends on pre-training a conditional diffusion model, which needs paired clean and degraded data. The clean data are used to build the loss function for supervision, while the degraded data act as conditioning inputs for the denoising network. Therefore, we use existing supervised datasets and additionally simulate a portion of the data to meet the requirements of mixed degradation. \
> Our method primarily addresses three common types of degradation in the fusion scenario: improper lighting, color distortion, and noise. For improper lighting, we use 2,220 image pairs from the MIT-Adobe FiveK Dataset [r1], covering images with varying exposures and their corresponding ground truth manually adjusted by photography experts. For color distortion, we use 1,031 image pairs from the Rendered WB dataset [r2], including color-biased images under various light sources such as fluorescent, incandescent, and daylight, as well as corresponding reference images manually calibrated under the Adobe standard. For noise, we add Gaussian noise, pulse noise, Poisson noise, Rayleigh noise, and uniform noise to 2,220 clean images from the MIT-Adobe FiveK Dataset and 2,220 clean images from the MSRS dataset to obtain noised images. All these image pairs constitute the complete dataset for training our diffusion model, driving our model’s learning for compound degradation removal.\
> [r1] Learning photographic global tonal adjustment with a database of input/output image pairs. CVPR 2011.\
> [r2] When color constancy goes wrong: Correcting improperly white-balanced images. CVPR, 2019.\
> [r3] PIAFusion: A progressive infrared and visible image fusion network based on illumination aware. Information Fusion, 2022.\
>
> Q2: Expand application scenarios.\
> Reply: This is an insightful suggestion. Of course, our proposed method can be further generalized to other multi-modal image fusion scenarios, as its methodology is a general fusion paradigm. To prove this, we conduct comparative experiments in the near-infrared and visible image fusion scenario. The visual results are shown in Fig. r5, where our method effectively integrates texture details from the near-infrared band with those from the visible image, while preserving natural color attributes of the visible image. Notably, the inherent image restoration capability of our method allows it to produce vivid fused images in underexposed scenes without causing overexposure like MRFS, as seen in the results of the first row. Furthermore, the quantitative results in Table r4 show that our proposed method ranks first in three of all five metrics and second in the other two. Overall, our method can be generalized to near-infrared and visible image fusion scenario with promising performance.
>
> Q3: Quantitative evaluation of polarization image fusion.\
> Reply: We conduct a quantitative assessment of the polarization image fusion, as reported in Table r4. Our method achieves the best scores in four of all five metrics, including EN, AG, SD, and SCD. These results indicate that our fused image contains the most information, the richest texture, the best contrast, and the most feature transfer from the source images. Overall, the quantitative results validate the advantages of our method in the polarization image fusion scenario, demonstrating its strong multi-modal fusion generalization capability.
>
> Q4: Brightness-chrominance separation.\
> Reply: Image fusion requires a high level of color fidelity to the scene. Taking the infrared and visible image fusion as an example, the colors in the fused image are required to be as consistent as possible with those in the visible image. Therefore, by independently purifying and preserving the chrominance components in the visible image, our method can effectively and conveniently achieve color fidelity.\
> Next, we discuss why our method does not directly process three-channel images. Firstly, from the perspective of image restoration alone, directly processing color images is entirely feasible. However, our method requires embedding information fusion into the latent layers of the diffusion model used for image restoration. This means that features from the gray infrared image could potentially interfere with the color distribution of features from the visible image. In particular, this interference occurs in the highly nonlinear latent space, where some small changes can be amplified by the decoder to produce large color distortions. In this case, ensuring the expected color fidelity is very difficult. Second, the interference is directly related to the way multi-modal features are fused. In our method, we use a nonlinear neural network called the Fusion Control Module to perform information aggregation, which is guided to retain significant thermal radiation objects while preserving rich background textures. These two goals correspond to the similarity loss functions (see Eqs. (9) and (10)) based on the indicators of pixel intensity and gradient. Under such optimization guidance, it is difficult to avoid disrupting the color distribution in the features from the visible image. For verifying, we adapt our proposed method to directly process three-channel images without separating brightness and chrominance components, and the results are presented in Fig. r1. Clearly, color distortion occurs. Furthermore, we implement quantitative evaluation in Table r1. The direct processing strategy decreases the color score CIECAM16 and also negatively affects other metrics to varying degrees
>
> Q5: Typos and underline on $AG$'s second place.\
> Reply: Thanks for pointing out these issues. We will carefully correct all typos in the final version and further improve the presentation of the figures and tables.

---

> > ### Comment · Reviewer_TtBn · 2024-08-10
> >
> > OK. Your reply solved my doubts.

---

> > > ### Author Response · Authors · 2024-08-13
> > >
> > > We truly appreciate your efforts in improving our paper. If this paper is accepted, we plan to incorporate the following revisions into the camera-ready version in response to your recommendations:
> > >
> > > 1. Supplement data construction details: We use existing supervised datasets (MIT-Adobe FiveK Dataset, Rendered WB dataset) and additionally simulate a portion of the data to meet the requirements of mixed degradation (such as improper lighting, color distortion, and noise). These data constitute the complete dataset for training our diffusion model, driving our model's learning for compound degradation removal.
> > >
> > > 2. Add additional application scenarios: We extend our proposed model to the near-infrared and visible image fusion task. The experimental results continued to demonstrate the advantages of our method.
> > >
> > > 3. Supplement quantitative evaluation of polarization image fusion: We supplement the quantitative assessment of the polarization image fusion, showing that our method achieves the best scores in four of all five metrics.
> > >
> > > 4. Discuss brightness-chrominance separation: We discuss the two reasons why our method does not directly handle three-channel images, including the nonlinear amplification of interference and the impact of fusion loss functions.
> > >
> > > We greatly appreciate your positive feedback on our work. If you have any further questions or concerns, please feel free to contact us.

---

### Official Review · Reviewer_DZyq · 2024-07-13

**Soundness:** 3
**Presentation:** 2
**Contribution:** 2
**Rating:** 6
**Confidence:** 2

**Summary:**

This paper proposes an interactive framework that can exploit the intrinsic connection between image restoration and multi-modal image fusion.
The authors embed information fusion within the diffusion process and address the "composite degradation challenge" i.e., multi-modal information integration with
effective information restoration from degradation like colour casts, noise, and improper lighting. Particularly, first, independent conditional diffusion models are applied
to each modality with compound degradation -- the degradation removal priors are embedded into the encoder-decoder network. A fusion control module (FCM) sits in
the multi-step diffusion process to manage the integration of multi-modal features and remove degradation during T-step sampling. Next, to interactively enhance
focus on objects of interest during diffusion fusion, the authors designed a text-controlled fusion re-modulation strategy that incorporates a text and a zero-shot OWL-ViT to
identify the objects of interest. In other words, this step performs a secondary modulation with the built-in prior to enhance saliency.

**Strengths:**

- It is interesting to see the effect of combining image restoration and multi-modal image fusion in a single framework.
 - The proposed method is well-motivated and the authors provide a clear explanation of the method.
 - The Text-controlled fusion re-modulation strategy could be useful in many applications.
 - The authors provide the code in the supplementary material (although I have only dry run the code and not tested it).
 - Extensive experiments are conducted to validate the proposed method.
 - The authors provide ablation studies to show the effectiveness of each component of the proposed method.

**Weaknesses:**

For now, I have minor concerns and mostly questions (as listed in the next section).
 - The authors should add a brief discussion on the competitors in supplementary material. For example, differences between TarDAL, DeFusion, LRRNet, DDFM, and MRFS.
 - Typo in Eq. 2: $\Theta_{t}^{X^{B}}$ should be $\Theta_{t}^{X^{b}}$.
 - Improve the caption of Figure 2. I had to read the entire paper to understand the figure (it should be self-explanatory).
 - Not much of a weakness, but the authors could improve the clarity of the paper if they added the tensor dimension of each variable in Figure 2.

**Questions:**

- In the proposed method, input visual image X is broken into brightness and chroma components. I wonder if this step is absolutely necessary -- or can we skip $\eta^{c}_{
    theta}$ and directly combine both $X$ and $Y$ as three-channel images $\mathbb{R}^{H \times W \times 3}$.
 - What if I use InstructIR (for image restoration) followed by MaxFusion (for multi-modal fusion) -- how would it compare with the proposed method?
    - (InstructIR) https://arxiv.org/pdf/2401.16468 | Github: https://github.com/mv-lab/InstructIR
    - (MaxFusion) https://arxiv.org/pdf/2404.09977 | Github: https://github.com/Nithin-GK/MaxFusion
 - In the Limitation and Future work section, will a no-training approach be possible? For example, "MaxFusion"-like approach but with the proposed deep integration of image restoration and multi-modal fusion.

**Limitations:**

The authors discussed limitation section in supplementary A.4 -- Particularly, Table S1 shows number of parameters and runtime. This is highly appreciated.

---

> ### Author Rebuttal · Authors · 2024-08-06
>
> Q1: Discussion on the necessity of the brightness-chrominance separation. \
> Reply: Unlike the image generation task emphasizing diversity, image fusion demands high color fidelity. For instance, in infrared and visible image fusion, the fused image should closely match the colors of the visible image. To this end, our method independently purifies and preserves the chrominance components of the visible image for color fidelity.\
> While a direct processing approach without separating brightness and chrominance is appealing, it faces several challenges in our work. First, from image restoration alone, directly processing color images by the diffusion model is entirely feasible. However, our method integrates information fusion into the latent layers of the diffusion model, which can cause gray infrared features to interfere with the visible image’s color distribution. Such interference in the nonlinear latent space can cause the decoder to amplify small changes into significant color distortions. Second, the interference is related to the feature fusion way. Our method uses a nonlinear neural network, the Fusion Control Module, to aggregate features. This module preserves key thermal objects and background textures, guided by similarity loss functions based on pixel intensity and gradient (see Eqs. (9) and (10)). This optimization can disrupt color distribution due to direct pixel changes. To verify, we adapt our method to process three-channel images without separating brightness and chrominance, as shown in Fig. r1. This strategy causes noticeable color distortion. We also conduct a quantitative evaluation. Table r1 reveals that the direct processing strategy lowers the color score CIECAM16 and negatively impacts other metrics to varying extents.
>
> Q2: InstructIR plus MaxFusion.\
> Reply: MaxFusion is designed for conditional image generation, requiring results to meet conditions like depth, skeleton, segmentation, and edges. Its strength lies in extending single-condition to multi-condition generation through feature fusion. However, MaxFusion is unsuitable for the image fusion task due to lower fidelity. Specifically, image fusion needs the fused image to maintain pixel-level consistency with multiple source images, preserving significant objects and sharp textures. In contrast, MaxFusion focuses on semantic consistency, producing diverse results with different styles. In Fig. r2, using MaxFusion for infrared and visible image fusion with infrared and visible images as conditions shows that it does not meet the goal of enhancing scene representation in image fusion.\
> Thus, we conduct comparisons using the reviewer-recommended InstructIR followed by several advanced image fusion methods. First, we input different text prompts into InstructIR to address improper lighting, noise, and color distortion. The restored images are then fused with advanced fusion methods. Results are shown in Fig. r3 and Table r2. Our method which implicitly integrates image restoration and fusion, shows better performance than these methods following a sequential strategy. In particular, our method can balance thermally salient object retaining and degradation removal, while competitors cannot.
>
> Q3: No-training approach.\
> Reply: MaxFusion extends single-modal image generation models to multi-modal ones through feature-level fusion. Specifically, MaxFusion proposes a no-training fusion strategy, which uses the variance maps of the intermediate feature maps of the diffusion model to select or weighted sum multi-modal features, assuming that pixels with higher variance represent higher priority for condition control.\
> Differently, our method uses a neural network, the fusion control module (FCM), which is trained under the constraints of Eq. (11) to fuse multi-modal features aiming at preserving significant objects and textures. Of course, our method can also be extended to a non-training version by using a statistics-based fusion strategy. In the ablation study, we have evaluated three no-training fusion strategies: maximum, addition, and mean (see Table 5).  These methods performed worse than our learnable FCM. Furthermore, we incorporate MaxFusion’s variance-based fusion strategy in our method (see Fig. r4 and Table r3), and it still falls short compared to our FCM. In future research, we will explore more powerful no-training fusion strategies, achieving performance comparable to retraining.
>
> Q4: Discussion on the competitors.\
> Reply: Due to limited space, we discuss only a few newer competitors. TarDAL uses a GAN with dual discriminators to preserve objects and textures, and collaborates with object detection for semantic optimization. DeFusion employs a masked autoencoder to decouple unique and common features, achieving complementary feature aggregation. LRRNet introduces low-rank representation model-based networks and combines pixel-level and feature-level losses for multi-modal fusion. MRFS couples image fusion and segmentation at the feature level to improve fused image quality. However, these methods do not fully address composite degradation, resulting in lower robustness. Given the diffusion model’s strong generative capabilities, applying it to image fusion could address composite degradation issues, but the lack of clean references complicates this. DDFM uses source images to guide sampling direction for implicit fusion, but still cannot solve composite degradation due to the inability to retrain the diffusion model. Our method tackles this by first optimizing the diffusion model and then embedding a fusion module within it, achieving integration of image restoration and fusion. Additionally, our method includes a text control interface for further modulation and enhancement of objects. These designs make our method both robust and flexible.
>
> Q4: Figures, tables, and typos.\
> Reply: We will carefully correct all typos in the final version and further improve the presentation of the figures and tables.

---

> > ### Author Response · Authors · 2024-08-13
> >
> > We thank Reviewer DZyq for the insightful and valuable feedback. We have included new experimental results to demonstrate the necessity of brightness-chrominance separation in our method. Additionally, we have conducted a comparison with InstructIR+Fusion to further highlight the advanced performance of our method. Furthermore, a no-training version of our methods has been explored. We hope we have addressed all of your concerns. If you have any additional questions, please let us know.

---

### Author Rebuttal · Authors · 2024-08-06

We sincerely thank each of the reviewers, area chairs, and program chairs for investing their time and effort into our paper. These valuable comments have enriched our understanding of the research problem and will greatly improve the quality of our manuscript. \
According to the reviewers' comments, we have added some validations. Firstly, we compare a three-channel direct processing strategy without brightness-chrominance separation, demonstrating the necessity of handling brightness and chrominance separately for color fidelity in our method. Secondly, we utilize the state-of-the-art image restoration method InstructIR as a preprocessing step of several advanced image fusion methods for further comparison, showing the advantages of our method with implicit integration of restoration and fusion. Thirdly, drawing inspiration from MaxFusion, we showcase the performance of our method with the non-training fusion strategy. Fourthly, we further extend the application scenario of our method to the near-infrared and visible image fusion, and supplement the quantitative results of the polarization image fusion. Fifthly, we validate the generalizability of the semantic gain offered by our proposed textual control in the object detection task. Finally, we investigate the impact of sampling steps on the final fused result and provide the basis for selecting 25 sampling steps in the experiments. All the figures and tables from the above validations are included in the global PDF file. In addition, we have prepared detailed individual responses for each reviewer to address and clarify all the raised issues and concerns.

---

### Decision · Program_Chairs · 2024-09-25

**Decision:**

Accept (spotlight)

**Comment:**

The authors have provided a good rebuttal. All four reviewers agreed that this paper proposed a novel and effective multi-modal image fusion method, and their concerns have also been properly resolved by the authors’ rebuttal. The final scores of this paper are all quite positive:  “Weak Accept”, “Weak Accept”, “Accept”, and “Accept”. The meta-reviewer agreed with the reviewers and recommended the acceptance of this paper.